# A Randomized Pilot Trial of Micronutrient Supplementation for Under-5 Children in an Urban Low-Cost Flat Community in Malaysia: A Framework for Community-Based Research Integration

**DOI:** 10.3390/ijerph192113878

**Published:** 2022-10-25

**Authors:** Crystal C. Wang, Muhammad Irfan Abdul Jalal, Zhi Liang Song, Yik Pheng Teo, Chin Aun Tan, Kai Voon Heng, Michelle Siu Yee Low, Azriyanti Anuar Zaini, Lucy Chai See Lum

**Affiliations:** 1Weill Cornell Medicine, 1300 York Avenue, New York, NY 10021, USA; 2UKM Medical Molecular Biology Institute (UMBI), Jalan Yaacob Latif, Bandar Tun Razak, Kuala Lumpur 56000, Malaysia; 3Department of Paediatrics, Universiti Malaya Medical Center, Kuala Lumpur 59100, Malaysia; 4Department of Occupational Safety & Health Unit, Hospital Tunku Azizah, Kuala Lumpur 50300, Malaysia; 5Department of Paediatrics, Hospital Tunku Azizah, Kuala Lumpur 50300, Malaysia

**Keywords:** child nutrition, pilot trial, stunting, micronutrient supplementation program, intervention program, urban poverty, Malaysia

## Abstract

Early childhood nutritional deficiency has detrimental consequences on physical and cognitive development. We conducted a single-center, single-blind, two-arm pilot randomized no-treatment controlled trial (the Child of Urban Poverty Iron Project (CUPIP); NCT03819530) in a people’s housing project locale in Selangor, Malaysia, between September 2019 and February 2020, to assess the trial’s general feasibility and preliminary benefits of daily micronutrient supplementation for iron storage and anthropometric outcomes in under-5 children. Those with history of premature births, congenital abnormalities, or baseline hemoglobin <70 g/L were excluded. Participants received baseline deworming and were simply randomized in a 1:1 ratio to either micronutrient (4-month daily micronutrient packets) or control (no micronutrient supplementation) groups. Information on anthropometric, erythrocytic, and iron storage endpoints were collected. Overall, 45 (25 micronutrient and 20 controls) participants were enrolled and completed 4-month endpoint assessments. Micronutrient recipients demonstrated higher median mean corpuscular volume, serum ferritin level with no significant differences in all anthropometric endpoints. In conclusion, this pilot trial was implementable, demonstrating that micronutrient supplementation significantly improved hematological, but not anthropometric, endpoints, of under-5-year-old children living in an underprivileged environment. A definitive well-designed trial with larger sample sizes and greater attrition control should be contemplated in the future.

## 1. Introduction

Maintaining a normal micronutrient profile during the first 1000 days of life, from the time of conception until the second year is critical for brain development, physical growth, and optimal immune system [1]. In developing nations, poor micronutrient status is attributed to several factors, including insufficient dietary uptake, infection, low quality diet, and substandard bioavailability of micronutrient-deficient food sources, or plant-based diets [2,3].

Micronutrients play great roles in normal physiological functions and child health. For instance, vitamin A is essential for maintaining normal immune function, epithelial tissue architecture, and retinal functions [4]. Moreover, vitamin C is required for collagen synthesis, facilitates the absorption of iron in the small intestine, and serves as a potent antioxidant that eliminates free radicals that may cause cellular damages [4]. Furthermore, micronutrients such as iron have critical function during oxidative phosphorylation that produces energy for diverse metabolic processes via adenosine triphosphate (ATP) formation [5]. Hence, the maintenance of normal micronutrient status in young children is of a paramount importance to ensure normal physical growth and cognitive development.

Micronutrient deficiency is not easily recognized clinically in the early stage, with devastating impacts on children’s wellbeing and lifelong ramifications such as impaired cognition, increased childhood mortality, stunting, and compromised immunobiological functions [6,7]. Micronutrient-deficient children are also more susceptible to sickness, poor performance in school, reduced productivity, and impaired intellectual and social development, which can affect future generations [8]. Furthermore, micronutrient deficiency has been linked with obesity and non-communicable disease [9,10]. Proper micronutrient intake during this time hence lays the foundation for optimal growth, health, and neurodevelopment across the lifespan.

Malaysia has experienced rapid urbanization in the past few decades. While Malaysia has done remarkably well in uplifting the citizens’ standard of living, there are emerging areas of concern, particularly regarding the wellbeing of the children of the urban poor. Though the prevalence of stunting has been declining globally, it is rising in Malaysia, even when compared to other countries that have comparable levels of income per capita [11].

The urban poor are housed in low-cost, government-subsidized housing known as Projek Perumahan Rakyat (PPR), or People’s Housing Projects, in and around Kuala Lumpur and other major cities [12]. A 2018 report by UNICEF revealed that 12% of children in these low-cost flats had less than 3 meals a day and 97% of households reported cost barriers in preparing healthy meals for their children [11]. One in two households reported that they did not have enough money to buy food in recent months and that malnutrition is a major concern for their children. This UNICEF report also found that children in Malaysia were among the most overweight and obese in the region. Among the low-cost flat residents, 22% of children under 5 years old were stunted, 15% underweight, 20% either wasted or severely wasted, and 23% either overweight or obese [11]. This was consistent with the 2016 National Health and Morbidity Survey, which reported 13.7% of children in Malaysia to be underweight and 20.7% to be stunted, with the highest prevalence among those aged 6–11 months and 24–35 months, respectively [13]. Additionally, the prevalence of anemia in Malaysia among preschool-aged children was estimated to be between 18% and 33% [14,15]. A follow-up survey in 2011 reported a similar picture, with 32% of children aged 6 months to 5 years old anemic and 0.1% of them severe anemic [15].

Alarmed by these statistics, we conducted a preliminary survey of the poorest of families in the bottom 40% (B40) income group of PPR Lembah Subang, one of the low-cost flat localities in Selangor state, and found that 33% of the children under 5 years old weighed below the 3rd percentile, consistent with the aforementioned UNICEF report [11]. These preliminary findings demonstrated the need for an intervention targeting the high prevalence of stunted and underweight children in this low-income community, particularly as its members also face other social vulnerabilities of low education, poor awareness in nutrition and child development, and domestic and economic instability.

Therefore, the aims of the Child of Urban Poverty Iron Project (CUPIP) trial are to investigate whether this trial is feasible for implementation and to assess the preliminary effects of micronutrient supplementation on erythrocytic and iron storage profiles and anthropometric measurements such as weight and height of the children living in a low-cost housing project in Malaysia. We therefore tested the null hypothesis that micronutrient supplementation would not significantly influence the changes in (i) erythrocytic parameters, (ii) iron storage parameters, and (iii) anthropometric endpoints. Figure 1 summarizes the conceptual framework of our trial.

## 2. Materials and Methods

### 2.1. Study Design

The CUPIP trial is a single-center, single-blind, two-parallel group, randomized no-treatment controlled trial conducted amongst children aged 6 months to 5 years residing at PPR Lembah Subang 1, located in the state of Selangor, Malaysia between October 2019 and February 2020 (4 months). The participants were randomly assigned in a 1:1 treatment-to-control allocation to receive, in addition to the baseline deworming treatment, either daily micronutrient packets for 4 months (intervention arm) or no micronutrient supplementation (control group). This residency area comprises a local population with mixed ethnic makeup and poor socioeconomic status (most are B40). This research article adheres to the 2010 Consolidated Standards of Reporting Trials (CONSORT 2010) extension for randomized pilot and feasibility trials and its extension statement, CONSERVE 2021, for reporting modified trials due to the global COVID-19 pandemic [16,17].

### 2.2. Eligibility Criteria

All children between the age of 6 months to 5 years residing at the designated study location (PPR Lembah Subang 1) were eligible for this study. Those with history of premature births, congenital abnormalities resulting in gross impairment, serum hemoglobin level less than 70 g/L at baseline, or parents who did not consent to trial participation were excluded from the trial.

### 2.3. Sampling Procedure

A purposive sampling method was used due to the paucity of eligible patients that attended full follow-ups due to the strict Movement Control Order (MCO) from March 2020 until October 2021 imposed by the Malaysian Government during the COVID-19 pandemic.

### 2.4. Randomization and Allocation Concealment

Children were randomly assigned to treatment or control groups in a 1:1 ratio using simple randomization without stratification based on a sequence of random numbers generated at a distant site by a researcher uninvolved in participant recruitment and endpoint assessment using the randomization subroutine of Microsoft Excel 2013 (Microsoft Corp, Redmond, WA, USA). Allocation concealment was guaranteed by ensuring that different study personnel carried out the randomization procedure, patient recruitment, and treatment assignment through a central randomization system.

### 2.5. Intervention Assignment and Blinding Status

We distributed flyers, posters, and leaflets with the help of community mothers in PPR Lembah Subang 1 to encourage participation. Registration, informed consent, anthropometric measurements, 24-h dietary logs, and blood for iron status were taken at baseline. All children received deworming treatment at baseline. 

The treatment group received baseline deworming medications (200 mg single-dose albendazole for <2-year-old participants and 400 mg single-dose albendazole for >2-year-old participants), 4-month supply of daily micronutrient supplementation manufactured by DSM Nutritional Products Ltd. (Kaiseraugst, Switzerland) and donated by DSM Nutritional Products Malaysia, nutritional supplement education (importance of adequate macro and micronutrients, consumption methods and necessity of compliance), and a compliance monitoring log. The full content of the micronutrient supplementation is given in Table 1 which is based on formulations used by previous studies [18,19], WHO/WFP/UNICEF’s joint statement [20] and joint recommendations by the WHO, ICCIDD and UNICEF [21]. 

The control group only received similar baseline deworming medications without any daily micronutrient supplementation. At 2 months following baseline assessment, a compliance check was conducted, and anthropometric measurements taken. At the 4-month visit, anthropometric measurements, 24-h dietary recall, and blood were taken. Only the outcome assessors were masked to the treatment status, while participants and research investigators were aware of the assigned intervention. Figure 2 summarizes the recruitment process and data collection timeline. 

### 2.6. Ethical Issues

The trial was conducted according to the Declaration of Helsinki, 1964, and International Conference on Harmonization Good Clinical Practice (ICH-GCP). Written informed consent was obtained from parents or guardians of each child on the day of trial registration (Appendix A). Hospital ethical review board approval was obtained from the University Malaya Medical Center (UMMC) Medical Research Ethics Committee (MREC) on 4 April 2019 (MREC ID No.: 2018105-6741). The trial has been registered with the Clinicaltrials.gov registry (ID: NCT03819530) and National Medical Research Registry (NMRR ID: NMRR-19-59-45609). 

### 2.7. Sample Size Calculation

The overall sample size for this trial was originally to be 300 (150 subjects per treatment arm) based on a moderate effect size (Cohen’s d = 0.5), standard deviation of 0.20 g/L for serum hemoglobin level (reference), type I error rate of 0.05 and 80% power. However, due to COVID-19 outbreak which resulted in substantial challenges to participant enrollment and follow-up completion, we modified the original design of the trial into a pilot trial, with the sample size specified according to Whitehead et al.’s [23] recommendations. Since we assumed small effect sizes (0.1  ≤  δ  <  0.3) for each study outcome, we set the sample size as 20 participants per groups so that 80% power could be achieved in the main trial that we will carry out in the future [23].

### 2.8. Primary and Secondary Outcomes

Primary endpoints assessed in this study are changes in serum ferritin and reticulocyte hemoglobin levels at 0 and 4 months after interventions, reflecting changes in iron status. Changes in serum C-reactive protein (CRP) level at 0 and 4 months after micronutrient supplementation is also another primary outcome of interest since ferritin is an acute phase reactant and therefore variability in serum ferritin level is influenced by inflammation induced by helminthic infections. Hence, serum CRP reflects the success of deworming treatment, and a low level of CRP signifies an absence of inflammation, allowing us to attribute any increases in serum ferritin levels to the success of micronutrient supplementation. The secondary endpoints are changes in age-and-sex standardized height and weight, after micronutrient supplementation. 

### 2.9. Recruitment Process

#### 2.9.1. Research and Community Teams

We recruited a group of research and community teams to assist with the study. The research team included the principal investigator (senior consultant), 2 research assistants (medical post-graduates) and volunteer physicians to aid in data and blood collection. 

Community volunteers were recruited from mothers living in the PPR area and subsequently underwent induction and training sessions. A total of 5 serial sessions was conducted for the volunteers to explain the research objectives, research timeline, volunteers’ role, and importance of early childhood nutrition. An open question and feedback session was conducted at the end of the training to clarify the doubts of volunteers. 

Project volunteers were recruited from medical and nursing students in the University Malaya, Faculty of Medicine, university students in Klang Valley, colleagues of the primary investigator, and medical doctors working at University Malaya Medical Center or Ministry of Health. Food bank volunteers also assisted during community days. 

The research core working team was then established, which included the project supervisor, 3 project managers, and 19 community volunteers. Project volunteers were recruited at community interactive sessions (campaigns, health screenings, supplement distribution and social program). 

We developed a training module for study personnel (research assistants and community mothers), carried out by 1 professor of pediatrics and 2 postgraduate research assistants. Medical student RAs were trained to collect pertinent child health and development background knowledge, to read and fill in data collection forms (Appendix A), consent forms, child growth charts, child 24-h dietary recall logs, and compliance charts, and handle distressed children during blood draws (distractions, play). Further training was conducted on how to ask dietary history questions and on general conduct when visiting participant homes. The community mothers were trained to monitor compliance of mineral supplements, understand best methods for administering mineral supplements specific to each age group, and record intercurrent illnesses and medications.

#### 2.9.2. Enrollment/Registration

The children recruited were given a baseline deworming medication (as part of the community program). The study group and participant ID number were recorded by an unmasked researcher. First, dietary log and demographic survey were administered and parent/family information were collected. Dietary log interviews were conducted by medical student RAs. The demographic surveys were self-administered, and checked for completion by an RA. The children had their first anthropometric measurements taken including height and weight. Blood was drawn by medical doctors only, with the help of medical student RAs. 

### 2.10. Intervention Administration

#### 2.10.1. Supplementation Procedure Demonstration (For Treatment Group Only)

We started our supplement distribution on 6 October 2019 to the intervention group. We reinforced their knowledge about this project, provided education (content of micronutrient packets, importance of adequate macro and micronutrients, consumption methods, and importance of compliance), and distributed a compliance monitoring log (Appendix A). Parents were advised on the use of daily supplementation, demonstrated by research personnel. Each child was advised to take 1 packet per day, administered with food or drink.

#### 2.10.2. Compliance Monitoring and Subsequent Visits

Compliance monitoring was organized by one of our project managers together with the community volunteers. Biweekly updates and feedback via phone call, text messages, or WhatsApp were directly obtained from the participants’ mother by our project manager. Monthly supplement distribution sessions were also organized by the core team. The compliance rate (in percentage) for each child, c_child_, was calculated using the following formula (1):(1)$$c_{\mathrm{child}}=\frac{n_{\mathrm{cst}}}{n_{\mathrm{whole}}}\times 100$$
where n_cst_ is the number of confirmed days when the daily micronutrient supplements were consumed by the child and n_whole_ is the total whole duration of trial in days. The team visited the treatment families biweekly and conducted log maintenance to record supplementation intake and took records of any associated symptoms/sicknesses. The control groups were also visited biweekly for documentation of symptoms/sicknesses. The community mothers worked in pairs and were matched with 1 medical student volunteer to aid in keeping records and data collection. Once a month, the research team met community mothers and distributed the next month’s supply of micronutrients and collected the past month’s record logs.

Apart from that, during these visits, we held interactive workshops on nutrition (food groups education, cooking classes), parenting, play, and hygiene/self-care (e.g., washing hands, brushing teeth, and free child haircut sessions) with the aid of different collaborative partners including student societies and NGOs.

### 2.11. Data Collection

We had 2 data collection time-points: during the recruitment phase and 4 months after the beginning of supplement distribution. During baseline visit and the first follow up session in February 2020, we conducted anthropometric measurement, 24-h dietary recall, and blood sample collection for both intervention and control groups. 

For the 24-h hour dietary recall log, mothers were asked to recall all the foods given to their child over the past 24 h, in chronological order. The consumed foods were then grouped based on their types: starchy foods, nuts and legumes, breast milk, formula milk, dairy products, animal protein, organ meat, eggs, orange-colored fruits and vegetables, green leafy vegetables, other types of vegetables, and sugary foods and drinks. 

For blood samples, hematological profile (serum hemoglobin, mean corpuscular volume (MCV)), plasma ferritin, and serum CRP were measured. Anemia is defined as serum hemoglobin concentration less than 110 g/L [24]. Anthropometric measurements were performed by trained medical student volunteers under the supervision of one research team personnel. These were done with the child in light clothes and no shoes using Portable SECA scales (SECA, Hanover, MD). Weight was measured using SECA 874 mobile flat scale accurate to 0.1 kg. Height was measured as the recumbent length for children up to 2 years of age using a SECA 417 mobile measuring board while the height of children > 2-years of age was measured using a SECA 217 stadiometer accurate to the nearest 0.1 cm. These measurements were subsequently converted into Z scores. 

Hematological, serum CRP and iron storage profiles, serum hemoglobin, hematocrit and MCV values, red cell and reticulocyte counts, reticulocyte-hemoglobin (RET-He), and others were analyzed using Sysmex XN-20 (Sysmex Corp., Kobe, Japan), plasma ferritin with Siemens ADVIA 2400 Chemistry System (Siemens Healthineers, Siemens Healthcare GmBH, Erlangen, Germany) and serum CRP with Siemens Centaur XP Immunoassay System (Siemens Healthineers, Siemens Healthcare GmBH, Erlangen, Germany). All measurements were conducted at the Clinical Diagnostic Laboratory of University Malaya Medical Center, Kuala Lumpur.

All data were recorded in paper-based clinical report forms (CRFs) and kept systemically in a filing system. We transferred the data into digital formats once the results for serum CRP, ferritin hemoglobin reticulocyte count results were obtained from the UMMC laboratory. The electronic data were password protected and kept in a locked cabinet located at the principal investigator’s office at the university. 

### 2.12. Statistical Analysis

The data were analyzed by using IBM SPSS Statistics for Windows, version 26 (IBM Corp., Armonk, NY, USA). The data were analyzed based on complete case analysis and no interim data analyses were carried out. Continuous data were summarized in mean/median and standard deviation/interquartile range whilst categorical data were summarized in frequency and percentage. The median differences in primary and secondary endpoints between the micronutrient and without-micronutrient groups after the second visit were compared using Mann–Whitney test. To compare the median differences in the primary and secondary outcome measures between pre-intervention and post-intervention periods, Wilcoxon signed rank tests were carried out. Fisher’s exact test or its extension, Fisher–Freeman–Halton test, (for a contingency table that was larger than a 2 × 2 contingency table)) was used [25,26,27] to compare the proportions of different dietary patterns (food groups) and categorical clinicodemographic profile (e.g., age, education level, antenatal complications, etc.) across treatment arms. McNemar’s tests (based on discordant pairs) were carried out to compare the dietary food groups before and after interventions were instituted, as stratified by the treatment arms. All tests were two-tailed and produced exact *p* values based on algorithms developed by Mehta and Patel [28]. The significant threshold was fixed at *p* < 0.05. 

## 3. Results

### 3.1. Recruitment and Registration

We registered a total of 168 children below 5 years old during the pre-launch and 2 days launch campaign. We also successfully provided deworming medication for all children from the community who attended the launch campaign, regardless of registration status. 

We reconfirmed interest of participation via phone call, text messages, and WhatsApp. In total, 84 children (50%; 95% CI: 42.5, 57.5) subsequently attended our official recruitment assessment, and we recruited children from 3 separate sessions between August and September 2019. In total, 81 subjects (48.2%; 95% CI: 40.5, 56.0) were randomized (41 controls and 40 micronutrient recipients). At the end of 4-month follow-up, 25 intervention (61.0%; 95% CI: 44.5, 75.4) and 20 control (50%; 95% CI: 35.2, 64.8) participants completed the whole assessments and were included in the analyses. The CONSORT flow diagram (Figure 3) summarizes the progress of our clinical trial. 

### 3.2. Baseline Characteristics

As shown in Table 2, both micronutrient and control groups have comparable baseline dietary patterns, clinical parameters, and demographic profiles. Nevertheless, the children in both groups were severely wasted at baseline as evident from median z-scores of less than −2.60 for the majority of anthropometric measurements (BMI and weight-to-height ratio). Maternal education level is proportionally at a lower level for the control groups, though the difference between the micronutrient and control arms is not statistically significant. No significant differences were found in the rest of other demographic profiles, dietary patterns, hematological and serum iron storage parameters. 

The prevalence of anemia at baseline is 14.0% (95% CI: 5.8, 28.6). Comparable baseline anemia prevalence was found in both groups (Control: 16.7% (95% CI: 5.5, 38.2); Micronutrient group: 10.5% (95% CI: 18.4, 34.5)).

### 3.3. Comparisons of Anthropometric, Erythrocytic, and Serum Iron Storage Profile at 4-Month Follow-Up

In Table 3, the medians for MCV and serum ferritin level are significantly higher in the micronutrient group. However, the inter-group differences in medians serum total iron biding capacity (TIBC), iron level and other anthropometric endpoints are not significant. No significant difference was found for other anthropometric, hematological, and serum iron storage endpoints. A significantly higher consumption of sugary foods was observed in the control group. Three participants experienced intercurrent illness (all upper respiratory tract infections (URTI)) in the micronutrient group.

In the micronutrient group, there was a high average compliance rate to daily micronutrient intake, based on compliance log monitoring (mean: 89.6%; SD: 10.3%; range: 63.00–100.00%). Only 1 participant experienced an adverse event (foul-smelling stool) in the micronutrient group. In both groups, no participants experienced serious adverse events (SAE) or suspected unexpected serious adverse reaction (SUSAR). 

### 3.4. Before-and-After Intervention Comparisons of Anthropometric, Erythrocytic, and Serum Iron Storage Profiles and Dietary Food Groups 

Table 4 shows the results when baseline and follow-up anthropometric and laboratory endpoints are compared separately for each group. The anthropometric outcomes improved in both control and micronutrient groups but more substantially and significantly for the control population. As for serum iron storage parameters, we found that the median serum ferritin level in the control group dropped from 25.6 to 21.0, whereas in the micronutrient group, it increased from 25.30 to 32.50 although the differences within the respective groups pre and post were both not significant. However, the TIBC level raised significantly in the control population which may indicate iron deficiency.

Based on pre- and post-intervention comparisons of food groups (Table 5), there are significant differences in terms of formula milk, eggs, and sugary consumptions only in the control group. Both groups demonstrated significant differences in pre- and post-intervention consumption of breast milk. Further analyses revealed non-significant differences with regard to the consumptions of the rest of other food groups.

## 4. Discussion

### 4.1. Micronutrient Supplementation and Trial Endpoints

Micronutrient deficiency has been implicated for a variety of illnesses such as iron-deficiency anemia, keratomalacia, xerophthalmia, elevated risk of respiratory infections, stunting, wasting, and cognitive underdevelopment in young children [29,30,31]. In the Malaysian setting, based on the findings of the Nutrition Survey of Malaysian Children (SEANUTS Malaysia) study, iron deficiency anemia, vitamin D deficiency, and to a lesser extent vitamin A deficiency were found the most common micronutrient deficiencies in children aged 6 months to 12 years [32]. To address this problem, there are a variety of methods to combat micronutrient deficiency among children implemented in diverse countries: food subsidies, distribution of fortified food, and micronutrient supplementation [33]. Many studies have shown that micronutrient supplementation has the highest efficacy among these 3 methods in addressing and curbing micronutrient depletion in children [34,35]. The findings from our pilot trial further support this approach, as this is the first-ever trial investigating the effects of micronutrient supplementation on children in a community experiencing urban poverty.

An interesting observation that we demonstrated in this pilot trial is the high proportion of under-5 children who were severely wasted. We hypothesized that this finding can be attributed to the severe micro- and macronutrient deficiency among participants in both treatment arms at baseline. At 4-month follow-up, we observed improvements in median z-scores for both BMI and weight-to-height ratio, which can be attributed to changes in dietary patterns secondary to nutritional education received by mothers of the children in both treatment groups and is evident in higher consumptions of formula milk, eggs and paradoxically sugary foods (Table 4). It is also additionally worth noting that micronutrient supplementation also resulted in greater improvements in BMI and weight-to-height ratio despite the non-significant pre-and-post intervention differences. Our findings corroborated the results of prior researchers who showed high proportions of wasting or severe wasting in under-5 children living in low-cost flats in Kuala Lumpur [11]. Hence, our results assert the importance of a nutritional intervention to address the undernutrition issues among children living in an urban poverty and deprivation setting. 

Further, it can be clearly observed that micronutrient supplementation improved several hematological markers and serum ferritin profile. Nonetheless, no significant differences were observed for any anthropometric endpoints between the micronutrient and control groups. Surprisingly, we found significant differences in the majority of anthropometric endpoints in the control group which can be attributed to significant increases in the intake of sugary food, eggs, and formula milk among the control participants at the 4-month follow up. Our findings thus parallel the observations made by Untoro et al. [36] who demonstrated that micronutrient supplementation improved micronutrient status and hematological profiles in 6-to-12-month-old Indonesian infants but did not improve their growth or morbidity. Our findings thus support the existing consensus among researchers that positive growth requires both micro and macronutrient supplementation [37,38]. However, our findings point towards a lack of hemoglobin recovery at 4 months post micronutrient supplementation, which parallels findings from previous research conducted among rural Mexican preschoolers [39]. Nonetheless, significant increases in percentage of reticulocytes observed in the intervention group based on a pre-and-post comparison demonstrated early erythropoietic recovery since improvement in reticulocyte counts precedes that of hemoglobin. In spite of this, longer periods of follow-up and micronutrient supplementation are required to conspicuously effect a recovery in hemoglobin level. 

We have demonstrated a positive effect of micronutrient supplementation on iron storage as evident from significantly higher median serum ferritin concentration in the micronutrient group at 4-month post supplementation, compared to a non-significant decrease in serum ferritin in the control group. This observation further supports the ideas that micronutrient supplementation enhances the iron storage profile and that deworming treatment alone was insufficient [40]. Moreover, the lack of significant effects of supplementation on hemoglobin levels may be explained by the shorter period of supplementation and our small sample size. In addition, we also omitted the RET-He variable from analysis due to the substantial number of missing data and the absence of estimates of local thalassemia prevalence, a known confounder for RET-He, in our study cohort which might lead to incorrect analysis and interpretation of results. To address this, we are currently exploring the relationships between RET-He and other serum iron storage parameters in our ongoing study. The findings will be subsequently used to evaluate the utility of RET-He as a primary trial endpoint for future definitive trials.

Other observations worth mentioning are the beneficial effects of deworming treatment observed in both control and micronutrient-supplemented groups and the possible synergism between deworming treatment and post-deworming micronutrient supplementation. Anthropometric endpoints and serum iron profile were generally improved in both groups, with much higher improvements documented in the micronutrient recipients, as evidenced by higher increases in serum ferritin, lower serum transferrin, and greater improvements in median weight and weight-to-height z scores. Paradoxically, the differences were only significant in the control group. We hypothesize that the much smaller sample size of the micronutrient group prevented the differences from being significant. Furthermore, soil-transmitted helminthiasis (STH) is considered to be endemic in Malaysia, but only among certain pockets of the population such as “Orang Asli” (indigenous people), refugees, and among dwellers of overcrowded urban centers with unsatisfactory household hygiene such as the PPRs [41]. Since STH is associated with iron deficiency anemia and overall nutritional deficits in pre-school children, we hypothesized deworming treatment alone would be inadequate to effect as much improvement as a combined deworming treatment augmented with micronutrient supplementation. 

However, Rajagopal, Hotez, and Bundy [42] argued in their review that the deworming and supplemented micronutrient cointervention was not recommended due to its ambiguous cost-effectiveness and laborious procedure, not because of its lack of effects. Higher improvements, albeit small and insignificant pre-and-post intervention differences, in hemoglobin, total red cell counts, serum iron, and ferritin profile in our deworming and micronutrient recipients support such arguments. Moreover, both groups had low median serum CRP and basophil and eosinophil percentages which suggests the success of deworming treatment in eradicating intestinal parasites that may interfere with optimal micronutrient absorption. Our findings are thus in tandem with Nga et al. [43], who demonstrated that micronutrient supplementation following a single albendazole administration resulted in higher plasma hemoglobin, ferritin, and iron concentrations than albendazole-alone recipients. 

We also observed that no participants in either group consumed internal organs which are rich in iron and other nutrients. We hypothesized that this might be a cultural issue or perhaps a particular deviation from traditional norms. As sources of animal proteins become cheaper, families may not maximize the value of available animal proteins, eating just the meat and discarding internal organs. Our postulation nevertheless requires further well-designed studies for verification.

Finally, scrutiny of our trial conduct and administration also revealed several insights that may further streamline the conduct of future definitive trials, most importantly, the substantial and variable amount of time, cost, and efforts required to recruit and follow up participants until trial completion, especially during the complete MCO (lockdown) phase due to the COVID-19 pandemic. We have thus meticulously recorded the challenges encountered which can be used for planning and organizing future definitive trials, specifically with respect to streamlining project timeline and allocating funding and research manpower to critical trial activities. This will prove an asset for future grant applications for similarly designed clinical trials that will be conducted in community-based research settings. Additionally, though our results have insufficient generalizability, a completely inevitable consequence due to the nature of our pilot trial, our overall trial design and data analysis techniques may serve as a blueprint for designing the methodology of future definitive trials with a longer follow-up period and analyzing the data accrued at each time point. In this case, our results obtained at the 4-month follow-up can guide future triallists to adopt the more flexible group sequential design (either frequentist or Bayesian approach) where the optimal number of interim analyses and precise stopping boundaries are required to be specified a priori [44,45]. The adoption of such group sequential design may result in a premature trial termination if the benefits of micronutrient supplementation are conspicuous even in the early period of trial [45,46]. Consequently, the findings of future trial results may be quickly utilized by relevant national policymakers and stakeholders to improve the nutritional profiles of under-5 children experiencing the deleterious effects of urban poverty and deprivation. 

### 4.2. Trial Limitations and Future Directions

One of the main limitations of our trial is the low sample size. This was inevitable due to the COVID-19 pandemic that severely hampered patient recruitment and follow-ups. As most families had informal employment, at the beginning of the COVID-19 pandemic, families lost jobs and went back to rural villages. Study subjects were thus lost to further follow-up. Moreover, this was the first clinical study carried out in this community, resulting in parental hesitancy in consenting to blood taking and further participant attrition and small sample size. We also did not administer any placebo intervention in the control group and hence the control participants could not be adequately masked which might result in expectation bias [47]. However, we believe this is minimal since objective trial endpoints were used instead of subjective trial outcomes such as self-reported quality of life or symptoms of micronutrient deficiency. Moreover, Staudacher et al. [47] also demonstrated the difficulties in designing and developing satisfactory sham interventions (placebo) in nutrition-related randomized controlled trials (RCTs), a cogent observation which can also be generalized to RCTs involving micronutrient supplementation. Nevertheless, based on the feedback received from the trial participants, we now have the relevant information for designing and developing satisfactory placebos (sham supplements) that have similar texture and tastes as the micronutrient supplement which can be used in future definitive (confirmatory) trials. This further emphasizes the values of a pilot (exploratory) RCT in improving the design of future definitive trials. In addition, we could not control the effects of confounders (e.g., differences in baseline hematological, iron, and serum CRP profiles) using a multivariable regression method since the sample size was too small and the normality assumption was violated for all outcome variables. Apart from this, the benefits of micronutrient supplementation on cognition (verbal and nonverbal intelligence) could not be sufficiently investigated due to the short follow-up period of our study. Hence, future studies should seek evidence on such associations and elaborate further on whether the cognitive benefits of micronutrient supplementation can be tempered by possible antagonistic effects of competitive uptake of micronutrients in the intestine, thus interfering with absorption and eventual bioavailability of one micronutrient relative to others.

Hence, a well-designed future definitive trial is required to further investigate our preliminary observations. We recommend that future definitive trials include a much bigger sample size. To aid in sample size calculation for the future definitive trial, we have provided additional parameter estimates such as mean and standard deviations for each trial endpoints (Appendix A). More precise trial designs, such as longer follow-up and micronutrient supplementation periods, double-blind design, use of random sampling prior to participant enrollment to minimize selection bias, implementation of intention to treat (ITT) analysis, and multiple imputation (MI) for missing data rather than as-treated analysis, can help minimize bias introduced by prognostic variable imbalances between the treatment arms [48,49,50,51]. For the last recommendation, we endeavored to carry out ITT analysis and multiple imputation, under missing at random assumption (MAR), for cases with missing information. This was unfortunately impossible due to the severity of missing data in our sample, small sample size, and the non-Gaussian (normal) nature of our data for the CUPIP trial’s endpoints, which prevented the MI algorithm from converging efficiently and accurately to plausibly true estimated effects, a known methodological limitation of MI [52,53]. If corrected, the actual effects of micronutrient supplementation can be ascertained more accurately. This evidence might then be used to effect national policy changes to improve the growth and quality of life of pre-school children coming from an underprivileged urban community (urban poor). 

## 5. Conclusions

Micronutrient supplementation is effective in improving several hematologic and iron outcomes, but not anthropometric endpoints, as evidenced from the significantly higher median serum ferritin concentrations and median MCV in the micronutrient group. Moreover, based on pre- and post-intervention comparisons, the median percentage of reticulocyte count is significantly higher in the micronutrient recipients than those in the control arms. Additional well-designed future definitive trials are nevertheless required to verify our findings and address our study limitations.

## Figures and Tables

**Figure 1 ijerph-19-13878-f001:**
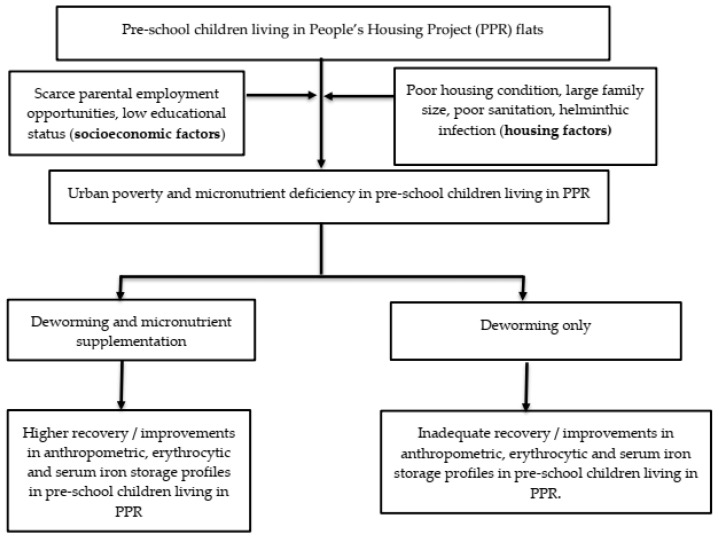
The conceptual framework of the CUPIP trial.

**Figure 2 ijerph-19-13878-f002:**
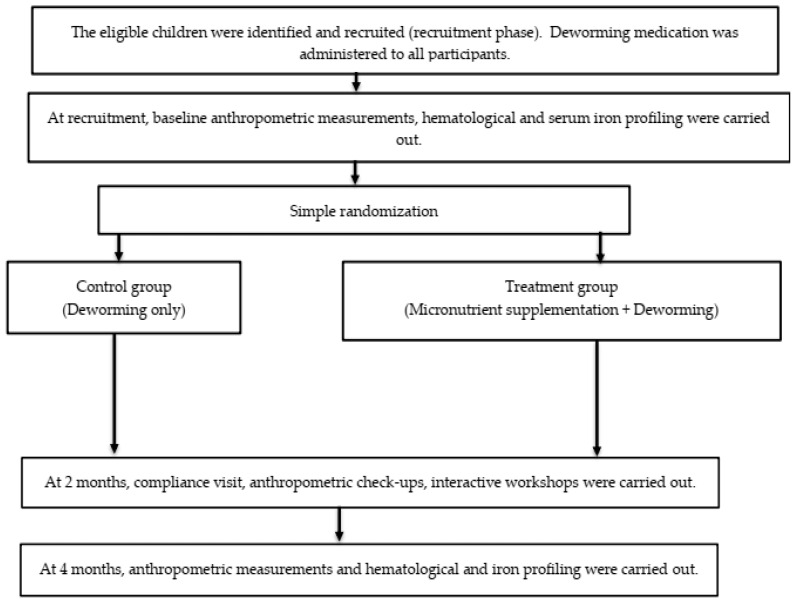
The flow of the study from recruitment until the completion of all follow-up visits.

**Figure 3 ijerph-19-13878-f003:**
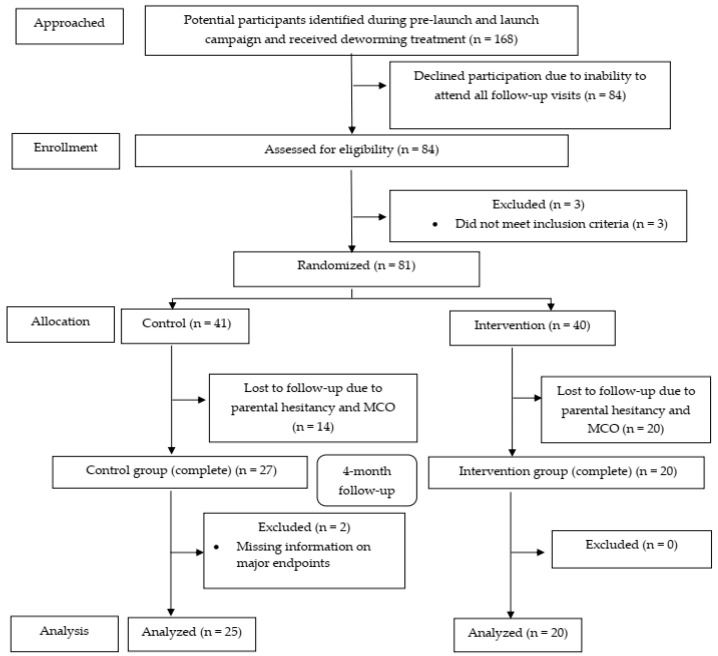
The CONSORT flow diagram of the study.

**Table 1 ijerph-19-13878-t001:** The full contents of micronutrient supplement (amount per 1 g sachet).

Nutrients	Nutrient Chemical Forms	Amount	% WHO-RNI ^a^ Daily Requirements [20,21,22]
Vitamin A	Retinyl Acetate	400 RE	100
Vitamin C	Ascorbic Acid	30 mg	100
Vitamin D	Cholecalciferol	5 µg	100
Vitamin E	dl-α-Tocopheryl Acetate	5 TE	100
Vitamin B1	Thiamine Mononitrate	0.5 mg	100
Vitamin B2	Riboflavin	0.5 mg	100
Vitamin B6	Pyridoxine	0.5 mg	100
Vitamin B12	Cyanocobalamin	0.9 µg	100
Folic acid	Folic Acid	90 µg	100
Niacin	Niacinamide	6 mg	100
Iron	Ferrous Fumarate	10 mg	100
Zinc	Zinc Gluconate	4.1 mg	100
Copper	Copper Sulfate, Anhydrous	560 µg	100
Iodine	Potassium Iodate	90 µg	100
Selenium	Sodium Selenite	17 µg	100

^a^ World Health Organization-Recommended Nutrient Intake.

**Table 2 ijerph-19-13878-t002:** Comparisons of baseline parameters between control and micronutrient groups (*n* = 45).

	Control (*n* = 25) *n* (%)	Micronutrient (*n* = 20) *n* (%) ^a^	*p*-Value ^b^
Median age for child (years) [IQR]	2.95 [2.14]	2.78 [2.64]	0.790
Gender (child)			
Male	13 (52.0)	9 (45.0)	0.767
Female	12 (48.0)	11 (55.0)	
Median maternal age [IQR]	35.00 [7.0]	32.00 [12.0]	0.448
Median maternal height [IQR]	155.5 [8.0]	154.25 [9.0]	0.974
Median paternal height [IQR]	165.00 [15.0]	170.00 [12.0]	0.516
Ethnicity			
Malay	16 (64.0)	14 (70.0)	0.369
Chinese	1 (4.0)	0 (0.0)	
Indian	7 (28.0)	3 (15.0)	
Others	1 (4.0)	3 (15.0)	
Maternal occupation			
Not working	15 (60.0)	16 (84.2)	0.100
Paid employment	8 (32.0)	1 (10.5)	
Self-employed	2 (8.0)	2 (5.3)	
Household income per month (in Ringgit Malaysia (RM))			
<3000	24 (96.0)	15 (78.9)	0.149
3000–6500	1 (4.0)	4 (21.1)	
Maternal highest education			
Primary School	6 (24.0)	0 (0)	0.066
Secondary School	16 (64.0)	16 (80.0)	
College	3 (12.0)	3 (15.8)	
Antenatal complication			
Yes	10 (40.0)	12 (63.2)	0.223
No	15 (60.0)	7 (36.8)	
Took iron supplements during pregnancy (mother)			
Yes	19 (76.0)	14 (73.7)	>0.999
No	6 (24.0)	5 (26.3)	
Taking iron supplement currently (mother)			
Yes	2 (8.0)	4 (21.1)	0.378
No	23 (92.0)	15 (78.9)	
Marital status			
Single	1 (4.0)	0 (0.0)	>0.999
Married	23 (92.0)	19 (100)	
Widowed	1 (4.0)	0 (0)	
Period of gestation			
<38 weeks	3 (12.0)	4 (21.1)	0.094
38–40 weeks	14 (56.0)	14 (73.7)	
>40 weeks	8 (32.0)	1 (5.3)	
Route of birth			
Vaginal delivery	16 (64.0)	13 (68.4)	>0.999
Caesarean section	9 (36.0)	6 (31.6)	
Feeding pattern in first 6 months			
Exclusively breastfed	14 (56.0)	11 (57.9)	0.634
Mixed breast & formula feed	10 (40.0)	6 (31.6)	
Exclusively formula fed	1 (4.0)	2 (10.5)	
Median volume of formula feed (ounce) [IQR]	4.00 [5.0]	4.50 [4.0]	0.743
Median frequency of formula feed (per day) [IQR]	4.00 [2.0]	3.00 [4.0]	0.541
Complementary food started			
Yes	25 (100.0)	19 (100)	-^c^
No	0 (0.0)	0 (0.0)	
Taking iron supplement currently (baby)			
Yes	1 (5.0)	3 (15.8)	0.300
No	24 (95.0)	16 (84.2)	
Starchy food			
Yes	25 (100.0)	20 (100.0)	-^c^
No	0 (0.0)	0 (0.0)	
Nuts and legumes			
Yes	3 (12.0)	5 (25.0)	0.435
No	22 (88.0)	15 (75.0)	
Breast milk			
Yes	24 (95.0)	18 (90.0)	0.577
No	1 (5.0)	2 (2.0)	
Formula milk			
Yes	11 (44.0)	8 (40.0)	>0.999
No	14 (56.0)	12 (60.0)	
Other dairy product			
Yes	17 (68.0)	15 (75.0)	0.745
No	8 (30.2)	5 (25.0)	
Animal protein			
Yes	21 (84.0)	19 (95.0)	0.362
No	4 (16.0)	1 (5.0)	
Organ meat			
Yes	0 (0.0)	0 (0.0)	- ^c^.
No	25 (100.0)	20 (100.0)	
Eggs			
Yes	10 (40.0)	12 (60.0)	0.236
No	15 (60.0)	8 (40.4)	
Orange pigmented fruits and vegetables			
Yes	8 (32.0)	5 (25.0)	
No	17 (68.0)	15 (75.0)	0.745
Green leafy vegetables			
Yes	8 (32.0)	7 (35.0)	>0.999
No	17 (68.0)	13 (65.0)	
Other fruits and vegetables			
Yes	13 (52.0)	8 (40.0)	0.550
No	12 (48.0)	12 (60.0)	
Sugary food			
Yes	7 (28.0)	6 (30.0)	>0.999
No	18 (72.0)	14 (70.0)	
Sugary drink			
Yes	14 (56.0)	10 (50.0)	
No	11 (44.0)	10 (50.0)	0.769
Sugary food and/or drink			
Yes	16 (64.0)	14 (70.0)	0.757
No	9 (36.0)	6 (30.0)	
Median height Z score [IQR]	−1.33 [1.46]	−0.88 [1.34]	0.395
Median weight Z score [IQR]	−2.67 [2.48]	−2.82 [2.22]	0.515
Median body mass index (BMI) Z score [IQR]	−2.80 [3.20]	−3.00 [4.98]	0.743
Median weight/height Z score [IQR]	−3.02 [2.58]	−3.67 [3.61]	0.911
Median hemoglobin (g/L) [IQR]	120.0 [13.0]	120.00 [9.0]	0.823
Median MCV (fl) [IQR]	73.00 [5.0]	75.0 [5.0]	0.292
Median reticulocyte (%) [IQR]	1.15 [0.53]	1.05 [0.37]	0.148
Median serum iron (umol/L) [IQR]	11.70 [6.20]	9.40 [6.02]	0.160
Median serum ferritin (ug/L) [IQR]	24.7 [25.3]	25.3 [32.8]	0.458
Median serum transferrin (g/L) [IQR]	2.40 [0.3]	2.55 [0.4]	0.134
Median total iron binding capacity (TIBC) (umol/L) [IQR]	60.2 [6.3]	64.1 [10.1]	0.134
Median serum C reactive protein (mg/L) [IQR]	0.05 [0.48]	0.04 [0.49]	0.929

^a^ Participant missing certain baseline clinicodemographic information (*n* = 1) was omitted during analyses for those particular variables. No participant was omitted for the analyses of trial endpoints; ^b^ Mann–Whitney or Fisher’s exact test; ^c^ Could not be computed.

**Table 3 ijerph-19-13878-t003:** Comparisons of dietary groups and clinical endpoints between control and micronutrient groups after 2nd follow-up (*n* = 45).

	Control ^a^ (*n* = 25) *n* (%)	Micronutrient (*n* = 20) *n* (%)	*p*-Value ^b^
Intercurrent illness			
Yes	0 (0.0)	3 (15.0)	0.080
No	25 (100.0)	17 (85.0)	
Starchy food			
Yes	19 (76.0)	20 (100.0)	-^c^
No	0 (0.0)	0 (0.0)	
Nuts and legumes			
Yes	6 (31.6)	11 (55.0)	0.200
No	13 (68.4)	9 (45.0)	
Breast milk			
Yes	0 (0.0)	1 (5.0)	>0.999
No	19 (100.0)	19 (95.0)	
Formula milk			
Yes	15 (78.9)	10 (50.0)	0.096
No	4 (21.1)	10 (50.0)	
Other dairy product			
Yes	15 (78.9)	12 (60.0)	0.301
No	4 (21.1)	8 (40.0)	
Animal protein			
Yes	15 (78.9)	17 (85.0)	0.695
No	4 (21.1)	3 (15.0)	
Organ meat			
Yes	0 (0.0)	0 (0.0)	-^c^
No	19 (100.0)	20 (100.0)	
Eggs			
Yes	15 (78.9)	13 (65.0)	0.480
No	4 (21.1)	7 (35.0)	
Orange pigmented fruits and vegetables			
Yes	8 (42.1)	8 (40.0)	>0.999
No	11 (57.9)	12 (60.0)	
Green leafy vegetables			
Yes	12 (63.2)	10 (50.0)	0.523
No	7 (36.8)	10 (50.0)	
Other fruits and vegetables			
Yes	10 (52.6)	10 (50.0)	>0.999
No	9 (47.4)	10 (50.0)	
Sugary food			
Yes	12 (63.2)	5 (25.0)	**0.025**
No	7 (36.8)	15 (75.0)	
Sugary drink			
Yes	15 (78.9)	12 (60.0)	0.301
No	4 (21.1)	8 (40.0)	
Sugary food and/or drink			
Yes	17 (89.5)	13 (65.0)	0.127
No	2 (10.5)	7 (35.0)	
Median height Z score [IQR]	−1.22 [1.57]	−1.12 [0.73]	0.428
Median weight Z score [IQR]	−2.05 [2.71]	−2.10 [2.77]	0.520
Median body mass index (BMI) Z score [IQR]	−2.10 [3.01]	−1.89 [3.59]	0.896
Median weight/height Z score [IQR]	−1.93 [2.95]	−2.11 [3.08]	0.801
Median hemoglobin (g/L) [IQR]	121.00 [11.0]	122.00 [14.3]	0.309
Median MCV (fl) [IQR]	75.00 [7.0]	77.50 [5.0]	**0.024**
Median reticulocyte (%) [IQR]	1.16 [0.45]	1.38 [0.73]	0.197
Median serum iron (umol/L) [IQR]	12.2 [7.6]	12.6 [5.9]	0.722
Median serum ferritin (ug/L) [IQR]	21.0 [22.9]	32.5 [26.4]	**0.007**
Median serum transferrin (g/L) [IQR]	2.60 [0.4]	2.40 [0.5]	0.120
Median total iron binding capacity (TIBC) (umol/L) [IQR]	65.30 [10.1]	60.20 [12.0]	0.120
Median serum C reactive protein (mg/L) [IQR]	0.07 [0.56]	0.09 [0.88]	0.672

^a^ Participants with missing variable information for dietary pattern (*n* = 6) were omitted during analyses; ^b^ Mann–Whitney (exact *p* value) or Fisher’s exact test.; ^c^ Could not be computed.

**Table 4 ijerph-19-13878-t004:** Comparisons of trial endpoints before and after administrations of interventions in each treatment arm (*n* = 45).

	Control (*n* = 25)			Micronutrient (*n* = 20)		
	Pre	Post	*p*-Value ^a^	Pre	Post	*p*-Value ^a^
Median height Z score [IQR]	−1.33 [1.46]	−1.22 [1.57]	0.650	−0.88 [1.34]	−1.12 [0.71]	0.552
Median weight Z score [IQR]	−2.67 [2.48]	−2.05 [2.71]	**0.042**	−2.82 [2.22]	−2.07 [2.77]	0.192
Median body mass index Z score [IQR]	−2.80 [3.20]	−2.10 [3.01]	**0.042**	−3.00 [4.98]	−1.89 [3.59]	0.261
Median weight/height Z score [IQR]	−3.02 [2.58]	−1.93 [2.95]	**0.036**	−3.67 [3.61]	−2.11 [3.08]	0.159
Median hemoglobin (g/L) [IQR]	120.0 [13.0]	121.0 [11.0]	0.242	120.0 [9.0]	122.00 [14.3]	0.652
Median MCV (fl) [IQR]	73.0 [5.3]	75.0 [6.5]	**0.001**	75.0 [5.0]	77.50 [4.5]	**<0.001**
Median reticulocyte (%) [IQR]	1.15 [0.53]	1.16 [0.45]	0.692	1.05 [0.37]	1.38 [0.73]	**0.021**
Median serum iron (umol/L) [IQR]	11.70 [6.20]	12.2 [7.6]	0.582	9.40 [6.02]	12.6 [5.9]	0.076
Median serum ferritin (ug/L) [IQR]	24.7 [25.3]	21.0 [22.9]	0.858	25.3 [32.8]	32.5 [26.4]	0.245
Median serum transferrin (g/L) [IQR]	2.40 [0.3]	2.60 [0.4]	**0.002**	2.55 [0.4]	2.40 [0.5]	0.106
Median total iron binding capacity (TIBC) (umol/L) [IQR]	60.2 [6.4]	65.30 [10.1]	**0.003**	64.1 [10.1]	60.20 [12.0]	0.103
Median serum CRP (mg/L) [IQR]	0.05 [0.48]	0.07 [0.56]	0.379	0.04 [0.49]	0.09 [0.88]	0.706

^a^ Based on Wilcoxon Signed-Rank Test with exact *p* value.

**Table 5 ijerph-19-13878-t005:** Comparisons of food groups, stratified by treatment arms, before and after intervention was instituted (*n* = 45).

	Treatment Groups	f_12_ ^a^	f_21_ ^b^	*p*-Value ^c^
Nuts and legumes	Control	2	6	0.289
	Micronutrient	2	8	0.055
Breast milk	Control	18	0	**<0.001**
	Micronutrient	17	0	**<0.001**
Formula milk	Control	0	7	**<0.001**
	Micronutrient	1	3	0.313
Animal protein	Control	2	1	>0.999
	Micronutrient	2	0	0.250
Eggs	Control	1	9	**0.021**
	Micronutrient	3	4	0.500
Orange pigmented fruits and vegetables	Control	3	8	0.227
	Micronutrient	4	7	0.274
Green leafy vegetables	Control	2	9	0.065
	Micronutrient	1	4	0.188
Other fruits and vegetables	Control	4	3	>0.999
	Micronutrient	3	5	0.363
Sugary food	Control	0	7	**0.016**
	Micronutrient	4	3	>0.999
Sugary drink	Control	3	6	0.508
	Micronutrient	5	7	0.387
Sugary food and/or drink	Control	1	5	0.219
	Micronutrient	5	4	0.500

^a^ The number of participants who consumed the food groups at baseline (pre-intervention) but stopped consuming at 4-month post-intervention follow-up (discordant pair 1); ^b^ The number of participants who did not consume the food groups at baseline (pre-intervention) but consumed it at 4-month post-intervention follow-up (discordant pair 2); ^c^ McNemar’s test (exact *p* value).

## Data Availability

The data used to corroborate the findings of this trial will be made available upon request to the corresponding author, L.L.C.S, at lumcs@ummc.edu.my.

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
