# Peer review of "A Randomized Pilot Trial of Micronutrient Supplementation for Under-5 Children in an Urban Low-Cost Flat Community in Malaysia: A Framework for Community-Based Research Integration"

_ijerph, 2022, doi:10.3390/ijerph192113878_

Round 1

Reviewer 1 Report

The work done on micronutrient supplementation in children is of great significance concerning to the public health of a country. The study is well-organized in terms of the scientific methodology used and data collection. I think the authors managed the limitations of COVID 19 also. However, I have some suggestions to be made in the manuscript prior to consideration in International Journal of Environmental Research and Public Health. My comments are provided below;

1. In the introduction it will good to include a paragraph on the role of micronutrients in child health and normal physiological functions

2. Regarding the inclusion and exclusion criteria used, I would like to know the BMI status of the participants. Whether the authors analyzed the BMI or body weight? Because it may also affect the data. Kindly incorporate the necessary details under the section 2.2

3. How the authors managed the diet of the participants, especially in treatment and control group? How you ensured that both these received similar nutrient content

4. What about the gender variation among the participants? Whether male and female participants are having similar micronutrient requirements?

5. The ill effects of micronutrient defficiency may be highlighted in the discussion section

6. The conclusion section could be improved by incorporating major finding

Author Response

1. In the introduction it will be good to include a paragraph on the role of micronutrients in child health and normal physiological functions

Response:

 We thank the reviewer for such a useful and pertinent feedback. We have incorporated a new paragraph that elucidates the role of micronutrients in child health and normal physiological functions (page 2, lines 47-55).

2. Regarding the inclusion and exclusion criteria used, I would like to know the BMI status of the participants. Whether the authors analyzed the BMI or body weight? Because it may also affect the data. Kindly incorporate the necessary details under the section 2.2

Response:

We thank the reviewer for his/her insightful comment. We had presented both baseline median age-and-sex-adjusted BMI Z scores ( i) median (IQR)CONTROL: -2.80 (3.20) ; ii) median (IQR) MICRONUTRIENT:-3.00 (4.98))  and age-and-sex-adjusted weight (BMI) z scores i) median (IQR)CONTROL: -2.67 (2.48); ii) median (IQR) MICRONUTRIENT:-2.82 (2.22))  in table 2 (page 14) and mean and standard deviations of both measures ( i)Age-and-sex adjusted mean BMI: Mean (SD)­CONTROL: -3.30 (3.13); Mean (SD)MICRONUTRIENT: -2.48(2.69)) in Table S1 (Supplementary Materials). Since the Z scores for baseline age-and-sex-adjusted BMI and weight are less than -2, we could confirm that the majority of children in both treatment arms were wasted. The Z scores for both measures were used, instead of the raw BMI or weight values, since BMI with varies with age and gender [1].

 We think that the information on both age-and-sex-adjusted BMI and weight Z scores should be incorporated in the result section since both measures are our main trial results (anthropometric component) and we did not exclude potential participants based on their baseline BMI or weight Z scores.

Reference

1.Flegal, K. M.; Ogden, C. L. Childhood Obesity: Are We All Speaking the Same Language? Advances in Nutrition 2011, 2 (2), 159S166S. https://doi.org/10.3945/an.111.000307

3. How the authors managed the diet of the participants, especially in treatment and control group? How you ensured that both these received similar nutrient content

Response:

The reviewer has raised a very valid and pertinent question. Since this trial was conducted in the actual pragmatic community setting, the diets of the participants could not be adequately controlled in the treatment and control groups, as evidenced from the higher proportions of dairy, milk and egg intake in the control group. We could not adjust the confounding effects of dietary intake differences between the individuals in the control and treatment arms using multiple linear regression approach due to our small sample size and the fact that our trial is exploratory and hypothesis generating, rather than explanatory and confirmatory, in nature. However, in our future definitive trial, the effectiveness micronutrient supplementation on the serum iron storage, anthropometric and hematological profiles of under-5 children living in an underprivileged urban setting will be thoroughly assessed, with statistical adjustments made to eliminate the confounding effects of different dietary intakes of participants using multiple linear regression technique.

4. What about the gender variation among the participants? Whether male and female participants are having similar micronutrient requirements?

Response:

We thank the reviewer for his/her insightful comments. The amount of each micronutrient provided in the supplement is more than sufficient to cater for daily micronutrient requirements for male and female children, as evidenced from previous research (refer table 2 and 3 of the reference) [1]. Besides, the distributions of female and male children are approximately similar in both micronutrient and control arms. Therefore, the gender-specific differences in micronutrient was not considered as a significant confounder whose effect needs to be adjusted for.

 References

  1. Cuadrado-Soto; Risica; Gans; Ellis; Araujo; Lofgren; Stowers; Tovar. Micronutrient Adequacy in Preschool Children Attending Family Child Care Homes. Nutrients2019, 11(9), 2134. https://doi.org/10.3390/nu11092134.

5. The ill effects of micronutrient deficiency may be highlighted in the discussion section

Response:

We thank the reviewer for the suggestion. We have amended our discussion section accordingly to include more details on the ill effects associated with micronutrient deficiency in children (page 20, lines 383-389).

6. The conclusion section could be improved by incorporating major finding

Response:

We again thank the reviewer for such pertinent recommendations. We have revised our conclusion based on the reviewer’s recommendations (page 23, lines 547-551).

Reviewer 2 Report

A Randomized Pilot Trial of Micronutrient Supplementation for Under-5 Children in an Urban Low-Cost Community in Malaysia: A Framework for Community-based Research Integration

 This manuscript is well organized and presents preparation procedures description for small preliminary study and its results that can help to organize full scale study how to provide brain health development important micronutrients supplementation for under-5 children. Publishing this manuscript can help other nutritionists well organize their studies.

I have no major comments. I did not find any great problem that would limit publishing this manuscript in specified journal. My comments are given in following list.

-          Lines 139-145 how it was check that children selected for treatment group they really took the prescribed doses of dietary supplements? I can see Material S4. In our nation we have clever sentence “Paper can handle anything”.

-          Lines 163-166 are based on reference Whitehead et al. Brackets [19] is placed at the end of line 166. I prefer to place brackets just after Whitehead et al.

-          Table 2 contains many numbers in [ ] brackets. It is not clearly explained what such numbers describe.

-          Page 11Table 2: parameter Gender: number for Micronutrient group Female is shifted to left to much.

-          Page 12 Table 2: parameter feeding has incorrectly given numbers (all are given in line valid for “Mixed breast and formula feed”.

-          Page 13 Table 2: parameter Animal protein: Yes and No should be shifted down one line. The same problem has to be solved for Green leafy vegetables and Other fruit and vegetables.

-          The same problem has to be solved in Table 3 on page 15 for Starchy food, Nuts and legumes and Formula milk.  The same problem can be seen on page 16 for Organ meat, Green leafy vegetables, Other fruit and vegetables, sugary food and Sugary drink.

-          Lines 402-405 place bracket with reference just after family reference Untoro et al. Your citation method may confuse the reader.

-          The same problem can be seen on lines 445-447. Please shift brackets behind “Bundy”.

-          The same problem can be seen on lines 453-456. Please shift brackets behind Nga et al.

-          Lines 580-582 reference seems me very complicated, several times replicates the same? Please, simplify it as possible.

-          Reference 39 need to shift one place left.

-          Material S2: are you sure that mothers can read and understand English? Line 50 needs to compress to left.

-          Material S3 Table ethnicity for mother´s has two fields for “Others Please specify:” The same problem can be seen part 5 Antenatal complications.

Author Response

1. Lines 139-145 how it was check that children selected for treatment group they really took the prescribed doses of dietary supplements? I can see Material S4. In our nation we have clever sentence “Paper can handle anything”.

Response:

We thank the reviewer for such comment. For the micronutrient group, the research team visited each participant’s household biweekly to assess the participant’s adherence to micronutrient supplementation and record any adverse effects associated with the micronutrient supplement. Similar frequency of household visits was also made for participants in the control group to record any illness / sickness. Besides, the community mothers, working in pairs and with 1 medical student volunteer, also helped with record keeping and documentation of treatment adherence. Further, the project manager contacted each participant’s mother biweekly through phone calls, Whatsapp and text messages to obtain additional information on the participant adherence to micronutrient intake.  Further information on compliance monitoring can be obtained in our manuscript (Section 2.10.2, page 8-9, line 232-247).

2. Lines 163-166 are based on reference Whitehead et al. Brackets [19] is placed at the end of line 166. I prefer to place brackets just after Whitehead et al.

Response:

The authors concurred with the recommendation made by the reviewers. The bracket with reference has been placed in the correct location as per reviewer’s suggestion.

3. Table 2 contains many numbers in [ ] brackets. It is not clearly explained what such numbers describe.

Response:

We thank the reviewer for the insightful comment. We have specified in the original table 2 within each row representing continuous numerical variables (e.g. median age of a child, median paternal and maternal ages, z score for BMI etc.) that the figures in the square bracket represent the interquartile range [IQR] for continuous numerical variable.

4. Page 11, Table 2: parameter Gender: number for Micronutrient group Female is shifted to left to much.

Response:

The authors concurred with the recommendation made by the reviewers. The number and percentage representing female participants in the micronutrient group has been shifted to the right place.

5. Page 12 Table 2: parameter feeding has incorrectly given numbers (all are given in line valid for “Mixed breast and formula feed”.

Response:

The authors concurred with the recommendation made by the reviewers. The number and percentage have been arranged according to the reviewer’s suggestion.

6. Page 13 Table 2: parameter Animal protein: Yes and No should be shifted down one line. The same problem has to be solved for Green leafy vegetables and Other fruit and vegetables.

Response:

Again, the authors concurred with the recommendation made by the reviewers. The number and percentage have been arranged according to the reviewer’s suggestion.

7. The same problem has to be solved in Table 3 on page 15 for Starchy food, Nuts and legumes and Formula milk.  The same problem can be seen on page 16 for Organ meat, Green leafy vegetables, Other fruit and vegetables, sugary food and Sugary drink

Response:

Again, the authors concurred with the recommendation made by the reviewers. The number and percentage have been arranged according to the reviewer’s suggestion.

8. Lines 402-405 place bracket with reference just after family reference Untoro et al. Your citation method may confuse the reader.

Response:

The authors concurred with the recommendation by the reviewers. The bracket with references has been placed in the correct location as per reviewer’s suggestion (page 20, line 409).

9. The same problem can be seen on lines 445-447. Please shift brackets behind “Bundy”.

Response:

The authors agreed with the reviewer’s suggestion. The bracket with reference has been shifted to the location as suggested by the author (page 21, line 452).

10. The same problem can be seen on lines 453-456. Please shift brackets behind Nga et al.

Response:

The authors agreed with the reviewer’s suggestion. The bracket with reference has been shifted to the location as suggested by the author (page 21, line 460).

11. Lines 580-582 reference seems me very complicated, several times replicates the same? Please, simplify it as possible.

 Response:

 We thank reviewer for the suggestion. The document was authored and published by the same organization, Institute of Public Health which is a sub-organization under the Malaysian NIH and Ministry of Health, hence the same name for the publisher and author.

12. Reference 39 need to shift one place left.

Response:

The authors agreed with the reviewer’s suggestion. Reference 39 had been shifted one place to the left (page 25, line 657).

13. Material S2: are you sure that mothers can read and understand English? Line 50 needs to compress to left.

Response:

The authors agreed with the reviewer’s suggestion. We have compressed line to the left. We also have provided the Malay version of the participant information sheet and consent form as appendix S3.

14. Material S3 Table ethnicity for mother´s has two fields for “Others Please specify:” The same problem can be seen part 5 Antenatal complications.

Response:

We thank the reviewer for his/her recommendation. However, both fields need to be retained to ensure correct information is recorded in the CRF. For mothers of different ethnicity or experiencing antenatal complications that are not specified in the CRF, the data collectors need to tick Others boxes first before specifying the ethnicity or specific antenatal complications.  Such practice is to ensure the integrity of information and facilitate the transfer of information from the paper-based CRF into the SPSS spreadsheet. This will ensure no information was incorrectly transferred to SPSS spreadsheet or being accidentally left out, resulting in high proportions of missing information for those particular variables.

Reviewer 3 Report

Some comments on the manuscript submitted for the review - many points need to be improved, for example:

- the study group was many times smaller than originally assumed,

- it is not clear why the authors used deworming and why with 1 dose,

- on what basis was the composition of the supplement developed?

- who made the supplement?

- what did the group receive as a placebo?

- why was the age discrepancy from 6 months to 5 years?

- in what chemical forms were the individual components? What% of the RDAs covered?

- why did the supplement provide just such ingredients, and not, for example, magnesium and calcium?

- it is necessary to plan this study well, and not future research based on it.

Author Response

1.The study group was many times smaller than originally assumed.

 Response:

We agree with the author’s comment that the actual number of participants recruited in this trial was many times smaller than what had been originally planned (n=300, NCT03819530). However, the original sample size was determined for a large-scale definitive trial, not for a pilot randomized trial.

Based on a pre-trial discussion with our research team members (research methodologist, statistician, People’s Housing Project community representatives, trial facilitators), we realized that critical information on the recruitment (enrolment) rate, retention (follow-up) rate, completeness of blood sample collection and barriers to trial implementation such as significant  number of individuals living in the People’s Housing Project community going back to their rural villages due to Covid-19 pandemic (page 22, line 423-424) and the acceptability of micronutrient supplementation had not been properly assessed.  Furthermore, such information and the estimates of trial endpoints (e.g. mean and standard deviations of serum iron storage profile, mean hemoglobin level etc) were also scarce when we scrutinized prior published literature. Hence, proper sample size calculation based on formal power analysis could not be carried out. As a result, based on recommendations from prior researchers [1-2], we decided to carry out a pilot randomized trial to assess the general technical feasibility of this trial and to obtain the relevant estimates for sample size calculation future definitive (confirmatory) trials.  This practice is in tandem with the recommendations from previous researchers that a well-conducted pilot randomized trial will provide useful information that will ensure smooth execution of future definitive trials [1-2].

There are several guidelines for determining the sample size for a pilot randomized trial [3-4]. We opted to use the recommendations by Whitehead et al since with a slightly larger sample size, the actual estimates of trial endpoints can be estimated more precisely in future definitive trials. Therefore, our choice of sample size for this pilot randomized trial is fully justified. Besides, a number of pilot randomized trials with small-to-moderate sample sizes have been published in this journal [5-7]. This further corroborates our choice of sample size for our study.

As a final point, it is worth mentioning that researchers should not be overly critical and dogmatize the sample size issue much, especially with respect to pilot randomized trials, since it can be a major hindrance to performing innovative translational research [8-9].

Reference

  1. Harvey, L. A. Feasibility and Pilot Studies Pave the Way for Definitive Trials. Spinal Cord2018, 56(8), 723–724. https://doi.org/10.1038/s41393-018-0184-x.
  2. Brown, S. M.; Paine, R.; Lanspa, M.; Gong, M. Value beyond the P: The Case for Higher-Quality and Better-Publicized Pilot and Feasibility Trials. Annals of the American Thoracic Society2019, 16(10), 1230–1233. https://doi.org/10.1513/annalsats.201901-059ps.
  3. Whitehead, A. L.; Julious, S. A.; Cooper, C. L.; Campbell, M. J. Estimating the Sample Size for a Pilot Randomised Trial to Minimise the Overall Trial Sample Size for the External Pilot and Main Trial for a Continuous Outcome Variable. Statistical Methods in Medical Research2015, 25(3), 1057–1073. https://doi.org/10.1177/0962280215588241.
  4. Julious, S. A. Sample Size of 12 per Group Rule of Thumb for a Pilot Study. Pharmaceutical Statistics2005, 4(4), 287–291. https://doi.org/10.1002/pst.185.
  5. Lederer, A.-K.; Storz, M. A.; Huber, R.; Hannibal, L.; Neumann, E. Plasma Leptin and Adiponectin after a 4-Week Vegan Diet: A Randomized-Controlled Pilot Trial in Healthy Participants. International Journal of Environmental Research and Public Health2022, 19(18), 11370. https://doi.org/10.3390/ijerph19181137
  6. Eibel, B.; Marques, J. R.; Dipp, T.; Waclawovsky, G.; Marschner, R. A.; Boll, L. C.; Kalil, R. A. K.; Lehnen, A. M.; Sales, A. R. K.; Irigoyen, M. C. C. Ventilatory Muscle Training for Early Cardiac Rehabilitation Improved Functional Capacity and Modulated Vascular Function of Individuals Undergoing Coronary Artery Bypass Grafting: Pilot Randomized Clinical Trial. International Journal of Environmental Research and Public Health2022, 19(15), 9340. https://doi.org/10.3390/ijerph19159340.
  7. Lozano-Berrio, V.; Alcobendas-Maestro, M.; Polonio-López, B.; Gil-Agudo, A.; de la Peña-González, A.; de los Reyes-Guzmán, A. The Impact of Robotic Therapy on the Self-Perception of Upper Limb Function in Cervical Spinal Cord Injury: A Pilot Randomized Controlled Trial. International Journal of Environmental Research and Public Health2022, 19(10), 6321. https://doi.org/10.3390/ijerph19106321
  8. Bacchetti, P.; Deeks, S. G.; McCune, J. M. Breaking Free of Sample Size Dogma to Perform Innovative Translational Research. Science Translational Medicine2011, 3(87), 87ps24–87ps24. https://doi.org/10.1126/scitranslmed.3001628.
  9. Bacchetti, P. Current Sample Size Conventions: Flaws, Harms, and Alternatives. BMC Medicine2010, 8(1), 17. https://doi.org/10.1186/1741-7015-8-17.

2. It is not clear why the authors used deworming and why with 1 dose,

 Response:

Deworming treatment was used to remove any lingering parasitic infection that may confound the effects of micronutrient supplementation on the trial endpoints. The intimate relationship between malnutrition and helminthic infections have been shown in the previous research [1].

For our research setting, it was found that parasitic infections were prevalent in children living in the urban slum area of Kuala Lumpur, Malaysia [2]. A single dose of 200 mg albendazole was used since it had been considered as a sufficient regimen for eradicating Ascaris Lumbricoides, Trichuris Trichuria and hookworm infections in previous research [3-4]. Further, single dose albendazole  had been found to be effective in reducing the rates of Trichuris trichiuria and Ascaris Lumbricoides  in early primary school children [5]. Therefore, it is critical to carry out a deworming therapy prior to micronutrient supplementation to assess the effect of micronutrient supplementation without being confounded by the presence of helminthic infections which on itself may cause iron-deficiency anemia in the trial participants [6].

References:

  1. Amare, B.; Moges, B.; Fantahun, B.; Tafess, K.; Woldeyohannes, D.; Yismaw, G.; Ayane, T.; Yabutani, T.; Mulu, A.; Ota, F.; Kassu, A. Micronutrient Levels and Nutritional Status of School Children Living in Northwest Ethiopia. Nutrition Journal2012, 11(1), 108. https://doi.org/10.1186/1475-2891-11-108.
  2. Bundy, D. A. P.; Kan, S. P.; Rose, R. Age-Related Prevalence, Intensity and Frequency Distribution of Gastrointestinal Helminth Infection in Urban Slum Children from Kuala Lumpur, Malaysia. Transactions of the Royal Society of Tropical Medicine and Hygiene1988, 82(2), 289–294. https://doi.org/10.1016/0035-9203(88)90450-6.
  3. Norhayati, M.; Oothuman, P.; Azizi, O.; Fatmah, M.S. Efficacy of single dose albendazole on the prevalence and intensity of infection of soil-transmitted helminths in Orang Asli children in Malaysia. Southeast Asian Journal of Tropical Medicine & Public Health. 1997 Sep;28(3):563-9.
  4. Nisha, M.; Faqiha, N.; Shafiqah, N.; Nurhidayah, S.; Davamani F. Efficacy of single dose albendazole treatment of soil-transmitted helminths among indigenous children in Malaysia. Southeast Asian Journal of Tropical Medicine & Public Health. 2021, 52(1), p.183-p.190.
  5. Raj, S. Mahendra.; Sein, K. T.; Anuar, A. Khairul.; Mustaffa, B. E. Effect of Intestinal Helminthiasis on School Attendance by Early Primary Schoolchildren. Transactions of the Royal Society of Tropical Medicine and Hygiene1997, 91(2), 131–132. https://doi.org/10.1016/s0035-9203(97)90196-6.
  6. Ngui, R.; Lim, Y. A. L.; Chong Kin, L.; Sek Chuen, C.; Jaffar, S. Association between Anaemia, Iron Deficiency Anaemia, Neglected Parasitic Infections and Socioeconomic Factors in Rural Children of West Malaysia. PLoS Neglected Tropical Diseases2012, 6(3), e1550. https://doi.org/10.1371/journal.pntd.0001550

3. On what basis was the composition of the supplement developed?

 Response:

The composition of the micronutrients in the supplement was based on the joint recommendations by the World Health Organization (WHO), World Food Program (WFP) and the United Nations Children’s Fund (UNICEF) for children aged 6 to 59 months (under-5 children)[1] and UNICEF/ International Council for Control of Iodine Deficiency Disorders (ICCIDD) and WHO’s recommended daily iodine intake in under-5 children [2-3].  These references have been included in the text (page, line)

References

  1. WHO, WFP, UNICEF. Joint Statement: Preventing and Controlling micronutrient deficiencies
    in populations affected by an emergency- Multiple vitamin and mineral supplements for pregnant and lactating women, and for children aged 6 to 59 months
    . WHO: Geneva, 2007.
  2. WHO, ICCIDD, UNICEF. Assessment of iodine deficiency disorders and monitoring their elimination. 3rd edition. WHO: Geneva, 2007: p.6.

4. Who made the supplement?

 Response:

The supplement was manufactured by DSM Nutritional Products Ltd (Kaiseraugst, Switzerland). The information had been included in the manuscript (page 4, lines 133-134).

5. What did the group receive as a placebo?

Response:

The participants in the control arm did not receive any placebo since this trial is a “a randomized no-treatment controlled pilot trial”, not “randomized placebo-controlled trial”. The reasons for this approach are two-fold:

1)  This is a pilot exploratory trial conducted in the actual community setting. The use of placebo is not universally and compulsorily warranted in pilot exploratory trials, as evidenced from prior studies that have similar trial design to ours [1-2]. In both studies, the participants in the control group received standard dietary advice and nutritional care [1-2]. We realized that the absence of placebo intervention in our trial may introduce expectation bias in the trial participants since they were not masked to the intervention status but we believe this effect is minimal since the trial endpoints were based on objective measurements of anthropometric, serum iron storage and hematological parameters, not on subjective outcomes (e.g. self-reported quality of life, symptoms of micronutrient deficiency etc.).

2) We could not find any suitable dietary means that might be adequately used as a placebo for the control arm due to the distinct and complex taste, texture and consistency of the micronutrient supplement. For instance, thiamine mononitrate (Vitamin B1) has slightly soupy taste whilst riboflavin (Vitamin B2) and copper have bitter and metallic tastes, respectively. Hence, the use of improperly-designed placebo would be futile in our setting since the trial participants would be able to distinguish the types of interventions received.

It is also worth mentioning that the design and development of satisfactory placebos for used in nutrition-related RCTs can be quite challenging. Staudacher et al. have shown that there are 9 criteria that need to be fulfilled for a sham diet to be considered as a satisfactory placebo in nutrition-related RCTs [3] and we believe these recommendations could also be generalized to RCTs involving micronutrient supplement. Based on our experience conducting this pilot exploratory RCT, criterion 1 (the content of the sham supplement must give the impression it is the true intervention supplement – modified from the original recommendation to suit our trial interventions) and criterion 9 (the sham supplement must not alter intake of other foods, food components or nutrients that might impact trial endpoints) are most pertinent and challenging to the development and design of satisfactory placebo for our trials. Nevertheless, based on the preliminary feedback from the trial participants, we have obtained sufficient information to design a satisfactory placebo that have similar tastes, forms and texture as the micronutrient supplement which can be used in future definitive (confirmatory) trials. This highlights the importance of a pilot (exploratory) randomized trial where the information obtained can be utilized to improve the design of future definitive (confirmatory) trials.

As a final note, we have incorporate the aforementioned points in our manuscript (page 22-23, line 516-528)

References

  1. Blancas-Sánchez, I. M.; Del Rosal Jurado, M.; Aparicio-Martínez, P.; Quintana Navarro, G.; Vaquero-Abellan, M.; Castro Jiménez, R. A.; Fonseca Pozo, F. J. A Mediterranean-Diet-Based Nutritional Intervention for Children with Prediabetes in a Rural Town: A Pilot Randomized Controlled Trial. Nutrients2022, 14(17), 3614. https://doi.org/10.3390/nu14173614.
  2. Patursson, P.; Møller, G.; Thomsen, B. B.; Olsen, E.; Mortensen, J.; Andorsdóttir, G.; Mohr, M.; Andersen, J. R. Effects of Postdischarge High-Protein Oral Nutritional Supplements and Resistance Training in Malnourished Surgical Patients: A Pilot Randomized Controlled Trial. Nutrients2022, 14(13), 2599. https://doi.org/10.3390/nu14132599.

      3.Staudacher, H. M.; Irving, P. M.; Lomer, M. C. E.; Whelan, K. The           Challenges of Control Groups, Placebos and Blinding in Clinical Trials of Dietary Interventions. Proceedings of the Nutrition Society 2017, 76 (3), 203–212. https://doi.org/10.1017/s0029665117000350.

6. Why was the age discrepancy from 6 months to 5 years?

Response:

We understand the reviewer’s concern about the wide age range of our study participants but our trial was concerned with addressing the prevalent micronutrient deficiency among under-5 children living in the urban low-cost flats in Malaysia, as evidenced from reference 9 in our manuscript (page 2, line 79). This is further supported by a recent systematic review that showed the positive effects of micronutrient supplementation on iron deficiency anemia, all-cause mortality and diarrhea in under-5 children [1]. Moreover, based on an estimate published by UNIFCE, micronutrient deficiency has affected 340 million under-5 children globally [2]. This underscores the importance of addressing micronutrient deficiency in children of this age group.

Reference

1.Tam; Keats; Rind; Das; Bhutta. Micronutrient Supplementation and Fortification Interventions on Health and Development Outcomes among Children Under-Five in Low- and Middle-Income Countries: A Systematic Review and Meta-Analysis. Nutrients 2020, 12 (2), 289. https://doi.org/10.3390/nu12020289.

  1. UNICEF. The State of the World Children 2019: Children, food and nutrition – Growing Well in a Changing World. UNICEF: New York, USA: p. 16.

7. In what chemical forms were the individual components? What % of the RDAs covered?

Response:

We thank the reviewers for their insightful comments. To address the issues raised by the reviewers, we have updated table 1 with two additional columns; i) Nutrient chemical forms; 2) WHO-RNI daily requirement.

8. Why did the supplement provide just such ingredients, and not, for example, magnesium and calcium?

Response:

Magnesium and calcium are macrominerals [1]. Our trial is solely concerned with the effects of micronutrient supplementation in the forms of trace minerals on iron storage, anthropometric and hematological parameters. Besides, based on a joint statement issued by the WHO, WFP and UNICEF, only 15 micronutrients were recommended to be included in micronutrient supplements / formulations [2]. Hence, magnesium and calcium are not included from the micronutrient formulation used in our trial.

Another point worth mentioning is we are currently investigating the effects of macrominerals and macronutrient supplementation on anthropometric, serum iron storage and hematological profiles in under-5 children experiencing urban poverty in our currently ongoing trial. The trial has a similar design to our CUPIP trial. The findings of this CUPIP trial are therefore beneficial in identifying the challenges associated with participant recruitment and follow-up and the overall trial conduct.

Reference

  1. Almeida, C. C.; Baião, D. dos S.; Rodrigues, P. de A.; Saint’Pierre, T. D.; Hauser-Davis, R. A.; Leandro, K. C.; Paschoalin, V. M. F.; Costa, M. P. da; Conte-Junior, C. A. Macrominerals and Trace Minerals in Commercial Infant Formulas Marketed in Brazil: Compliance with Established Minimum and Maximum Requirements, Label Statements, and Estimated Daily Intake. Frontiers in Nutrition2022, 9, 857698. https://doi.org/10.3389/fnut.2022.857698.
  2. WHO, WFP, UNICEF. Joint Statement: Preventing and Controlling micronutrient deficiencies in populations affected by an emergency- Multiple vitamin and mineral supplements for pregnant and lactating women, and for children aged 6 to 59 months. WHO: Geneva, 2007.

9. It is necessary to plan this study well, and not future research based on it.

Response:

We totally comprehend the reviewer’s concern but the smooth execution of future definitive trials is dependent on the findings of pilot trials to assess the general feasibility of definitive future trials, to estimate the recruitment rate and retention rate of an intervention, to identify potential challenges associated with the conduct of future definitive trials, to calculate the actual sample size required for future definitive trials and others and to familiarize the researchers with the clinical interventions that will be used in future definitive trials [1-2]. Conducting a large-scale definitive multi-centre clinical trial prematurely, without high-quality pilot or feasibility trials being carried out first, would result in inconclusive definitive trial results [2].  To aid researchers to decide whether a future definitive trial is feasible based on the findings of a pilot trial, Mellor et al. proposed that a set of progression criteria based on four domains (i)process -recruitment / attrition rates, intervention fidelity etc.; ii) resources – appropriate research venue, equipment and resources reliability etc.; iii) management – time to ethics approval at each trial site, rate of missing data etc.; iv) scientific-safety and adverse events, intervention tolerability, effect size etc.) should be pre-specified to conclude the feasibility of future definitive trials [3].

As an example, a number of high-quality pilot trials with small-to-moderate sample size (n=  15 to 53 study participants)  have been recently published in the International Journal of Environmental Research and Public Health [4-6]. In those examples, the study authors identified potential challenges associated with their respective trials (i.e. insufficient follow-up time, single centre trials, difficulties in accessing samples etc.) and the preliminary estimates of their respective trial endpoints. All of them concluded that the preliminary effects of the interventions found during their pilot randomized trials should be replicated in large-scale future definitive trials. This again emphasizes our belief that the design of future definitive trials should be based on the information gained and challenges identified during the earlier pilot randomized trials.

As a conclusion, a well-conducted pilot trial will generate invaluable information to ensure the smooth conduct of future definitive trials that will verify the preliminary findings of a pilot trial.

Reference

  1. Harvey, L. A. Feasibility and Pilot Studies Pave the Way for Definitive Trials. Spinal Cord2018, 56(8), 723–724. https://doi.org/10.1038/s41393-018-0184-x.
  2. Brown, S. M.; Paine, R.; Lanspa, M.; Gong, M. Value beyond the P: The Case for Higher-Quality and Better-Publicized Pilot and Feasibility Trials. Annals of the American Thoracic Society2019, 16(10), 1230–1233. https://doi.org/10.1513/annalsats.201901-059ps.
  3. Mellor, K.; Eddy, S.; Peckham, N.; Bond, C. M.; Campbell, M. J.; Lancaster, G. A.; Thabane, L.; Eldridge, S. M.; Dutton, S. J.; Hopewell, S. Progression from External Pilot to Definitive Randomised Controlled Trial: A Methodological Review of Progression Criteria Reporting. BMJ Open2021, 11(6), e048178. https://doi.org/10.1136/bmjopen-2020-048178.
  4. Lederer, A.-K.; Storz, M. A.; Huber, R.; Hannibal, L.; Neumann, E. Plasma Leptin and Adiponectin after a 4-Week Vegan Diet: A Randomized-Controlled Pilot Trial in Healthy Participants. International Journal of Environmental Research and Public Health2022, 19(18), 11370. https://doi.org/10.3390/ijerph19181137
  5. Eibel, B.; Marques, J. R.; Dipp, T.; Waclawovsky, G.; Marschner, R. A.; Boll, L. C.; Kalil, R. A. K.; Lehnen, A. M.; Sales, A. R. K.; Irigoyen, M. C. C. Ventilatory Muscle Training for Early Cardiac Rehabilitation Improved Functional Capacity and Modulated Vascular Function of Individuals Undergoing Coronary Artery Bypass Grafting: Pilot Randomized Clinical Trial. International Journal of Environmental Research and Public Health2022, 19(15), 9340. https://doi.org/10.3390/ijerph19159340.
  6. Lozano-Berrio, V.; Alcobendas-Maestro, M.; Polonio-López, B.; Gil-Agudo, A.; de la Peña-González, A.; de los Reyes-Guzmán, A. The Impact of Robotic Therapy on the Self-Perception of Upper Limb Function in Cervical Spinal Cord Injury: A Pilot Randomized Controlled Trial. International Journal of Environmental Research and Public Health2022, 19(10), 6321. https://doi.org/10.3390/ijerph19106321

Round 2

Reviewer 1 Report

No more comments to the authors

Reviewer 3 Report

The answers provided by the authors are exhaustive and explain the disputed issues. I think it is necessary to correct all the formatting so that the manuscript complies with the MDPI guidelines.